# Outcomes of Ixazomib Treatment in Relapsed and Refractory Multiple Myeloma: Insights from Croatian Cooperative Group for Hematologic Diseases (KROHEM)

**DOI:** 10.3390/medicina60111905

**Published:** 2024-11-20

**Authors:** Josip Batinić, Barbara Dreta, Goran Rinčić, Antonia Mrdeža, Karla Mišura Jakobac, Delfa Radić Krišto, Milan Vujčić, Mario Piršić, Željko Jonjić, Vlatka Periša, Jasminka Sinčić Petričević, Božena Coha, Hrvoje Holik, Toni Valković, Marija Stanić, Ivan Krečak, Ante Stojanović, Domagoj Sajfert, Sandra Bašić-Kinda

**Affiliations:** 1Division of Hematology, Department of Internal Medicine, University Hospital Center Zagreb, 10000 Zagreb, Croatia; 2School of Medicine, University of Zagreb, 10000 Zagreb, Croatia; 3Division of Hematology, Sisters of Charity Hospital, 10000 Zagreb, Croatia; 4Division of Hematology, Clinical Hospital Merkur, 10000 Zagreb, Croatia; 5Division of Hematology, University Hospital Split, 21000 Split, Croatia; 6Division of Hematology, University Hospital Dubrava, 10000 Zagreb, Croatia; 7Division of Hematology, University Hospital Centre Osijek, 31000 Osijek, Croatia; 8Department of Internal Medicine, General Hospital “Dr. Josip Benčević”, 35000 Slavonski Brod, Croatia; 9Department of Internal Medicine, Specialty Hospital Medico, 51000 Rijeka, Croatia; 10Division of Hematology, Department of Internal Medicine, Clinical Hospital Centre Rijeka, 51000 Rijeka, Croatia; 11Department of Internal Medicine, General Hospital Šibenik, 22000 Šibenik, Croatia; 12School of Medicine, University of Rijeka, 51000 Rijeka, Croatia; 13University of Applied Sciences, 22000 Sibenik, Croatia

**Keywords:** ixazomib, lenalidomide, multiple myeloma, relapsed/refractory, retrospective study

## Abstract

*Background and Objectives*: Ixazomib, used in combination with lenalidomide and dexamethasone (IRd), has shown efficacy in clinical trials for relapsed/refractory multiple myeloma (RRMM). *Materials and Methods*: This study evaluates the real-world effectiveness and safety of IRd in Croatian RRMM patients. A retrospective analysis was conducted on 164 RRMM patients treated with ixazomib at nine Croatian haematology centres from November 2016 to February 2023. Data on patient demographics, treatment regimens, and outcomes were collected and analysed using Kaplan–Meier survival curves and Cox proportional hazards models in R. The median age at ixazomib initiation was 66 years (range 40–91). *Results*: The overall response rate (ORR) was 65.8%, with 42% of patients achieving a very good partial response (VGPR) or better. The median progression-free survival (PFS) was 15.4 months, while median overall survival (OS) was 28.2 months. Hematologic toxicities included anaemia (53%), neutropenia (50%), and thrombocytopenia (45%). Infective complications, primarily COVID-19 and pneumonia, were reported in 38% of patients. The safety profile was consistent with previous studies, indicating manageable adverse events. Ixazomib-based therapy is effective and well tolerated in a real-world Croatian RRMM population. *Conclusions*: The findings align with clinical trial results, demonstrating the applicability of ixazomib in routine clinical practice. Further studies are needed to optimise treatment sequencing and improve patient outcomes.

## 1. Introduction

In recent decades, many new efficacious therapies were approved for patients with both newly diagnosed (NDMM) and relapse/refractory multiple myeloma (RRMM) [1,2]. Some of these new options are combinations which include ixazomib, the first proteasome inhibitor taken orally [3]. Based on the results of TOURMALINE-MM1 phase III clinical trial, the combination of ixazomib, lenalidomide, and dexamethasone (IRd) was approved for RRMM patients in the EU and the USA [4]. The TOURMALINE-MM1 study included patients who had received at least one prior line of therapy (median 1 prior line of treatment, range 1–3) and the results showed a significant improvement in median progression-free survival (PFS) using IRd compared to placebo-Rd (20.6 versus 14.7 months; hazard ratio [HR] 0.74; *p* = 0.01). There was an improvement in overall response rate (ORR), with no additional toxicity and a preserved level of quality of life [5]. As with all treatment options, the decision to use IRd is made by considering several factors: exposure to drugs (lenalidomide and/or bortezomib), efficacy, toxicity, and patient characteristics (age, comorbidities, frailty, cytogenetic abnormalities). This is particularly important in elderly patients, for whom frailty and comorbidity can be major limiting factors in the choice of therapy [6,7,8,9]. In that context, all oral combination with good safety profiles, such as Ird, can be used. However, some reports estimate that a large proportion of typical RRMM patients (approximately 40%) do not meet inclusion criteria in clinical trials, which leads to discrepancies between results reported in clinical trials and results reported in real-life practice [9,10,11]. As a consequence, it is uncertain whether we can translate results from clinical trials to real-life practice. Thus, real-world studies are needed to better describe real-life populations and to define optimal treatment sequencing for each patient, taking into account patients’ characteristics [8,12,13,14,15]. The objective of this non-interventional retrospective study is to evaluate ixazomib use in real life in the Croatian RRMM population.

## 2. Materials and Methods

We performed a retrospective analysis of RRMM patients with measurable disease who were treated with ixazomib (at least 1 cycle) in 9 Croatian haematology centres in the period between November 2016 and February 2023. Ixazomib was first made available as a patient–name programme in 2016 and purchases have been reimbursed by health care authorities since June 2019. A total of 164 patients with RRMM were included who received at least one prior line of therapy. All patients who received ixazomib-based therapy were enrolled in this study. The inclusion criteria were as follows: patients with RRMM, aged ≥ 18 years, who had received at least one prior line of therapy. Patients’ characteristics are shown in Table 1. The diagnosis of MM and response criteria was performed according to the International Myeloma Working Group [16]. We defined cases of high-risk cytogenetics as having one or more of the following cytogenetic and/or FISH analysis abnormalities: del17p; translocations—t(4;14) and t(14;20); and the duplication/amplification of chromosome 1 and complex karyotype.

This study was conducted in accordance with the ethical principles of the Declaration of Helsinki [17].

### Statistics

The statistical analysis for this study was conducted using the R programming language, leveraging its robust statistical and graphical capabilities. The primary endpoint of the study, progression-free survival (PFS), was analysed using Kaplan–Meier survival curves. PFS was measured as the time from start date of ixazomib-based therapy until the date when disease progression was documented. Disease evaluation was performed after every 2–4 cycles of therapy (evaluation after every 4 cycles is mandated by health care authorities). The survival package in R was utilised to fit survival models and estimate median survival times. The survfit function was employed to generate Kaplan–Meier plots, and the log-rank test was used to compare survival curves between different patient subgroups. We statistically compared subgroups according to the number of previous lines of therapy; age, with 70 years of age being the cut-off limit; and ECOG performance status.

## 3. Results

Data on a total of 164 patients with RRMM treated with ixazomib were analysed. The median age at the start of ixazomib treatment was 66 years (range 40–91). There group comprised 44% males and 56% females. The median number of previous lines of therapies was 1 (range 2–8). The majority of patients (134) were treated with a combination of ixazomib, lenalidomide, and dexamethasone (IRd), while the rest (30) were treated with other combinations (ixazomib with dexamethasone alone; ixazomib with melphalane and dexamethasone; ixazomib with cyclophosphamide and dexamethasone; Ixa + VAD; Ixa + Benda). Overall, 155 patients (94%) were exposed to bortezomib and 50 (30%) were exposed to lenalidomide. Only 10 (6%), 19 (12%), and 7 (4%) patients were exposed to carfilzomib, daratumumab, and pomalidomide, respectively. Overall, 50 patients (30%) had previously undergone autologous stem cell transplantation (ASCT) in the first line. Overall, 65% of patients had a performance status of 0–1 and 35% had a status ≥ 2, according to the Eastern Cooperative Oncology Group (ECOG). The overall response rate (better or equal to partial response; PR) for the whole group was 65.8%. A very good partial response (VGPR) or better was achieved in 42% of patients (Table 2). The median follow-up was 14.6 months and the median progression-free survival (PFS) was 15.4 months (at 12 months, the median PFS was not reached in 59% of cases), as shown in Figure 1a. The median overall survival (OS) was 28.2 months (at 12 months, the median OS was not reached in 72% of cases), as shown in Figure 1b. Regarding the number of previous lines of treatment, the median PFS was 15.4 months for patients treated with ixazomib in the second line, 17.3 months for those treated in the third line, and 11.5 months for patients treated in the fourth and subsequent lines of therapy (Figure 2a). The difference was not statistically significant. There was also no difference in OS regarding the number of previous lines of therapy (Figure 2b). The median PFS according to age was 14.7 months for the age group ≤ 70 years and 20.1 months for the age group ≥ 71 years, which reached statistical significance (*p* = 0.045, Figure 3a). There was no difference in OS in the specified age groups (Figure 3b). An age of 70 years was chosen as the cut-off value because, if the cohort was stratified into more age groups, statistical significance was not reached, probably due to small numbers of patients in each group. When stratifying patients according to ECOG performance status, the median PFS was 11.8 months in the ECOG 0 group, 16 months in the ECOG 1 group, 15.8 months in the ECOG 2 group, 18 months in the ECOG 3 group, and 15 months in the ECOG 4 group (Figure 4a). There was no statistical significance. The median OS values were as follows: 15.8 months in the ECOG 0 group, 30 months in the ECOG 1 group, 23 months in the ECOG 2 and 3 groups, and 15 months in the ECOG 4 group, with no statistical significance (Figure 4b). When analysing the group of patients who received IRd (134 patients), both the median PFS and OS were similar to those of the entire group: the median PFS was 15.4 months, and the median OS was 28.5 months (the median follow-up was 14.9 months). There were only 18 patients with high-risk cytogenetics (del17p—8 patients; t(4;14)—1 patient; chromosome 1 duplication—3 patients; complex karyotype—6 patients). In this small group of patients, the median PFS was 16.1 months, and median OS was 29.4 months. Unfortunately, for the majority of patients, data on cytogenetics and FISH analysis are missing or unknown. We also preformed ANOVA analysis to reduce potential biases. In the ANOVA analysis, we included the factors of prior ASCT, ISS stage, age and sex. There was no statistical significance regarding PFS and prior ASCT (F = 0.115; *p* = 0.735), ISS stage (F = 0.136; *p* = 0.713), age (F = 0.81; *p* = 0.37), and sex (F = 3.452; *p* = 0.065). We only found statistical significance regarding the OS and ISS stage (F = 3.59; *p* = 0.059), while other parameters did not show statistical significance: prior ASCT (F = 2.38; *p* = 0.125), age (F = 0.739; *p* = 0.391), and sex (F = 1.006; *p* = 0.317).

Anaemia, neutropenia, and thrombocytopenia were reported in 53%, 50%, and 45% of patients, respectively (Table 2). Infective complications were reported in 38% of patients. The most common infections were COVID-19 and pneumonia (Table 2).

During follow-up, 101 (62%) patients experienced disease progression, and 85 patients died (52%). The cause of death was disease progression in 22% of patients, followed by infective complications in 36% of patients. The cause of death was not determined in 32% of patients but was not disease-related.

## 4. Discussion

The use of ixazomib, in combination with lenalidomide and dexamethasone (IRd), for the treatment of relapsed and refractory multiple myeloma (RRMM) has been well documented in clinical trials such as TOURMALINE-MM1. However, translating these results into real-world settings can often present challenges due to differences in patient populations, health care settings, and treatment protocols. This study aimed to bridge this gap by evaluating the real-world efficacy and safety of ixazomib in Croatian patients with RRMM. Our study included a diverse population of 164 patients, treated across nine Croatian haematology centres, providing a broad perspective on the application of ixazomib in everyday clinical practice. The median age at the start of ixazomib treatment was 66 years, which is consistent with the pivotal study of TOURMALINE-MM1, where the median age was also around 66 years [4], while in the other real-world data reports such as the work of Furlan et al. [18] and Macro et al. [19], the median age was a higher (72.5 and 72 years, respectively). In our cohort, the overall response rate (ORR) was 65.8%, with 42% of patients achieving a very good partial response (VGPR) or better. This aligns with the findings of the TOURMALINE-MM1 trial, which reported an ORR of 78% and a VGPR rate of 48% or better [4]. These slight differences may be attributed to the broader inclusion criteria and the heterogeneous nature of the real-world population, which often includes patients with more comorbidities and different prior treatment exposures. When compared to the study by Macro et al. [19], our ORR was slightly lower. Macro et al. reported an ORR of 71% in their real-world analysis of ixazomib-based therapy [19]. However, their cohort included a higher proportion of patients treated in earlier lines of therapy, which may explain the higher response rate. In contrast, our study population had a higher median number of prior therapies, which is typically associated with lower response rates. Furlan et al. [18] conducted a similar study in an Italian population, reporting a median PFS of 17.2 months and an ORR of 69%. Our median PFS of 15.4 months was slightly lower, which could be attributed to differences in patient demographics and treatment protocols. Furlan et al. [18] also observed a high incidence of hematologic toxicities, similar to our findings, underscoring the importance of monitoring and managing these adverse events in clinical practice. Hajek et al. [20], in their analysis of the TOURMALINE-MM1 trial, emphasized the consistent benefit of using ixazomib across various subgroups, including patients with high-risk cytogenetic abnormalities. Our study corroborates these findings, demonstrating that the benefits of ixazomib-based therapy extend to real-world populations, including those with high-risk features. However, the slightly lower PFS in our study may reflect the challenges of managing a more heterogeneous patient population in routine clinical settings (patients with one or more comorbidities, a higher ECOG status, and older age). On a similar note, the results of the survival analysis of groups according to ECOG performance status in our study are interesting since better results were achieved in higher-category groups, although there was no statistical significance between outcomes. This can be explained by the fact that these frail patients were also older, and thus more likely to have received fewer prior lines of therapies, and the majority of them received ixazomib as a second line of treatment. This is also reflected in survival analysis according to age: a better PFS was achieved in older patients, a finding which was statistically significant (*p* = 0.045). This can also be explained by the fact that older patients received fewer prior lines of therapy and received ixazomib in the second line of treatment. Regarding non-haematological toxicities, we noticed that in our study there were greater numbers of infective complications than in the pivotal study. One of the explanations could be the Sars-CoV2 pandemic, since the majority of infective complications were Sars-CoV2 infections. Also, in our group, we reported no gastrointestinal adverse effects. We can only speculate that these adverse effects were mild and were not reported. Since IRd is one of the protocols used for the treatment of RRMM, we were interested in comparing these results with the outcomes of RRMM patients treated with other commonly used protocols: daratumumab in combination with lenalidomide, dexamethasone (DRd) and karfilzomib, lenalidomide, and dexamethasone (KRd). For that purpose, we analysed Croatian patients treated with the abovementioned protocols. In patients treated with DRd, after a median follow-up of 15.2 months, the median PFS was 24 months and the median OS was 24 months. In patients treated with KRd, after a median follow-up of 11 months, the median PFS was 13.6 months and median OS was 25 months. According to this preliminary analysis, we can say that our real-world data show similar outcomes in the IRd- and KRd-treated RRMM patients and somewhat better outcomes in patients treated with DRd, although these three groups were not compared directly. Parts of the data sets on KRd and DRd were presented as poster presentations at the 20th International Myeloma Society Annual Meeting in 2023 [21,22]. More elaborate analysis of both DRd- and KRd-treated RRMM patients is being prepared for publication.

This study has several strengths, including the provision of real-world data that offer valuable insights into the practical application of ixazomib in a diverse patient population. The inclusion of multiple haematology centres ensures a broad representation of clinical practice across Croatia, enhancing the generalizability of our findings. Additionally, comparing our results with those of established clinical trials and other real-world studies highlights the external validity and applicability of ixazomib-based therapy in routine practice. However, our study also has limitations, one of most obvious being the rather small number of patients. The retrospective design may introduce selection bias and limits our ability to establish causality. Furthermore, the median follow-up period of 14.6 months may not capture long-term outcomes and late-onset adverse events. The diverse patient characteristics, including varying lines of prior therapy and comorbidities, introduce variability that may affect the generalizability of our findings.

## 5. Conclusions

Our study demonstrates that ixazomib, used in combination with lenalidomide, dexamethasone, or other options, is an effective and well-tolerated treatment for relapsed/refractory multiple myeloma in a real-world Croatian population. The findings are consistent with clinical trials, underscoring the applicability of ixazomib-based therapy in routine practice. However, further prospective studies with longer follow-up times are needed to optimise treatment strategies and improve patient outcomes.

## Figures and Tables

**Figure 1 medicina-60-01905-f001:**
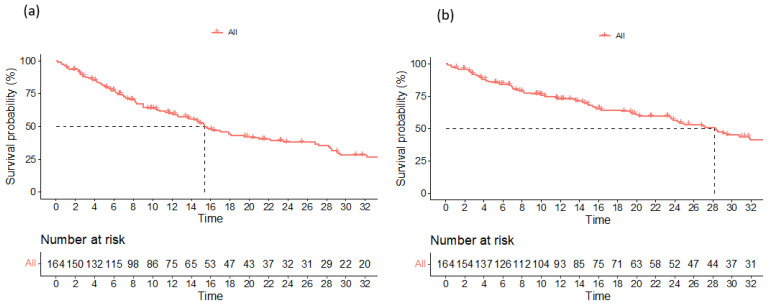
Progression-free survival (**a**) and overall survival (**b**) for the entire study population.

**Figure 2 medicina-60-01905-f002:**
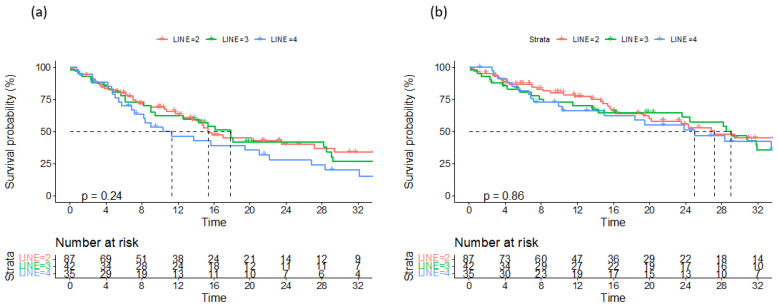
Progression-free survival (**a**) and overall survival (**b**) according to number of previous lines of treatment.

**Figure 3 medicina-60-01905-f003:**
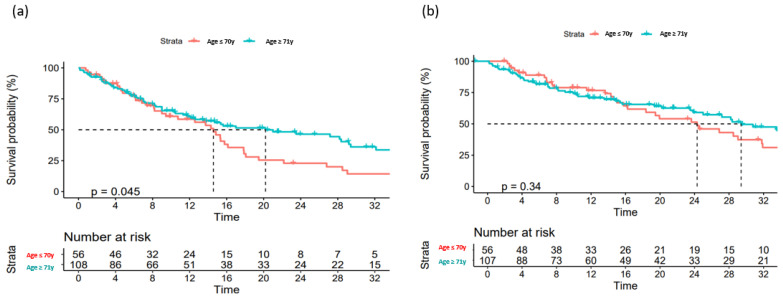
Overall survival (**a**) and progression-free survival (**b**) according to age.

**Figure 4 medicina-60-01905-f004:**
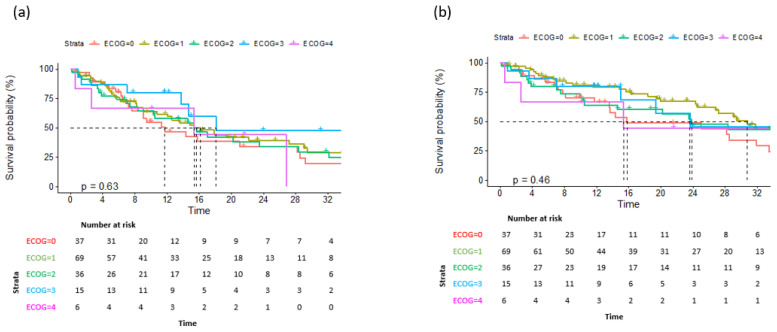
Overall survival (**a**) and progression-free survival (**b**) according to ECOG performance status.

**Table 1 medicina-60-01905-t001:** Study population characteristics.

Characteristics	Study Population (N 164)
Age (years) - at diagnosis; median (range) - at ixazomib start; median (range)	60 (33–87) 66 (40–91)
Sex—male; number (%)	72 (44%)
Preexisting comorbidities - cardiovascular - pulmonary - renal insufficiency - gastrointestinal - diabetes - neurological - other malignancies (prior to MM diagnosis) *	105 (64%) 12 (7%) 46 (28%) 8 (5%) 29 (18%) 10 (6%) 10 (6%)
ECOG performance status; number (%) - 0 - 1 - 2 - 3 - 4	38 (23%) 69 (42%) 36 (22%) 15 (9%) 6 (4%)
ISS stage at study entry; no. (%) - I - II - III	160 (100%) 42 (26%) 61 (38%) 57 (36%)
Cytogenetics—high risk; no. (%) Del17p t(4;14) chromosome 1 duplication complex karyotype	18 (11%) 8 1 3 6
Number of prior lines of therapy; median (range) - one prior line; no. (%) - two prior lines; no. (%) - ≥three prior lines; no. (%)	2 (1–8) 87 (53%) 43 (26%) 34 (21%)
Number of cycles received; median (range)	8 (1–59)
Previous exposure; no. (%) - bortezomib - carfilzomib - lenalidomide -pomalidomide - daratumumab Previous ASCT; no. (%)	155 (94%) 10 (6%) 50 (30%) 7 (4%) 19 (12%) 50 (30%)

ECOG—Eastern Cooperative Oncology Group; ISS—international staging system; Del17p—deletion of short arm of chromosome 7; complex karyotype—two or more cytogenetic abnormalities on conventional metaphase cytogentics; ASCT—autologous stem cell transplantation. * overall, 3 patients had colorectal cancer; 2 had breast cancer; 2 had non-Hodgkins lymphoma; 1 had a myeloprolipherative neoplasm; 1 had prostate cancer; and 1 laringeal cancer.

**Table 2 medicina-60-01905-t002:** Overall response rates and adverse events.

Overall response; n (%) CR VGPR PR MR SD	30 (18%) 39 (24%) 39 (24%) 1 (1%) 22 (13%)
Haematologic toxicity	
Anaemia, n (%) Grade 1 Grade 2 Grade 3 Grade 4	162 (100) 44 (27) 27 (17) 13 (8) 1 (1)
Thrombocytopenia; n (%) Grade 1 Grade 2 Grade 3 Grade 4	160 (100) 31 (20) 15 (9) 22 (14) 3 (2)
Neutropenia; n (%) Grade 1 Grade 2 Grade 3 Grade 4	160 (100) 48 (30) 14 (9) 14 (9) 3 (2)
Adverse events	
Infective complications COVID-19 Pneumonia Bronchitis Acute respiratory infection Urinary tract infection GI tract infection - C. difficile - Salmonella	No. of patients 10 10 1 2 4 3 1 1

CR—complete response; VGPR—very good partial response; PR—partial response; MR—minimal response; SD—stabile disease.

## Data Availability

The data presented in this study are available on request from the corresponding author due to privacy concerns.

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
