# Peer review of "Outcomes of Ixazomib Treatment in Relapsed and Refractory Multiple Myeloma: Insights from Croatian Cooperative Group for Hematologic Diseases (KROHEM)"

_medicina, 2024, doi:10.3390/medicina60111905_

Round 1

Reviewer 1 Report

Comments and Suggestions for Authors

1. Table 1 says complex karyogram and according to what I reviewed, what is globally accepted is complex karyotyp

2. describe abbreviations in table footers, for example ASCT CR VGPR PR MR SD etc.

3. I think that the material and methods should be described as to how many centers participated since it is a very interesting multicenter work and the data is clear.

Reviewer 2 Report

Comments and Suggestions for Authors

Dear Authors, 

Your findings are in agreement with other published studies. The reported work is deemed to be of interest due to the increase in the global incidence of frailty in the population.

However, the are some concerns regarding the description  of the methodology used and important details are not provided.

Materials and Methods

Authors should report if patients were consecutively enrolled and the eligibility/exclusion criteria. Moreover, clinical disease characteristics of patients are poorly described.

Acronyms should be expanded the first time they are mentioned, for example SD in table 2.

In Table 1, the median of number of prior lines of therapy is 1 with a range of 2-8. Please revise.

Statistics section

Please define the progression-free survival  measured and the time-frame when the assessment of the response to treatment was performed.

It is not clear if survival differences were statistically compared in subgroups.

Results section

In the sentence “at 12 months OS was not reached” do the authors  intend “at 12 months median OS was not reached”?

Define Infective complications as adverse events rather than toxicities.

The authors should explain the choice of 70 years as cut-off in the OS analysis.

Discussion section

You should compare the efficacy and safety of ixazomib with other RRMM treatment protocols.

Since clinical characteristic of the enrolled patients  are not well described, some of the  conclusions are non supported by the results.

Please check the references. Some of  them are not correct. Thus,  it is not possible for the reviewers to verify the conclusion.

Comments on the Quality of English Language

Minor revision is required.

Reviewer 3 Report

Comments and Suggestions for Authors

In this manuscript "Outcomes of Ixazomib Treatment in Relapsed and Refractory Multiple Myeloma: Insights from Croatian Cooperative Group
for Hematologic Diseases (KROHEM)
", a team led by Josip and Sandra, performed a retorspective analysis based on 164 RRMM patients treated with ixazomib, to evalutate the efficicy of ixazomib - based therapy. 

1. No significant difference observed in the study. Current data may not solidly support the conclusion.

2. What's the different between ixazomib combination group (134 patients) and the rest of them (30 patients) in overall survival and progression free survival.

3. ANOVA analysis might be used to decrease the risk of bias. 30% of the patients received ASCT, dose this will affect the evaluation of ixazomib-based therapy? and other factors, such as the ISS stage, cytogenetics, sex, age, also needs to included in ANOVA analysis.

Round 2

Reviewer 3 Report

Comments and Suggestions for Authors

The author addressed most of the comments in this revised version. The manuscript is much improved and can be accepted on the current version.